# Effects of a High Fat Meal Associated with Water, Juice, or Champagne Consumption on Endothelial Function and Markers of Oxidative Stress and Inflammation in Young, Healthy Subjects

**DOI:** 10.3390/jcm8060859

**Published:** 2019-06-15

**Authors:** Olivier Rouyer, Cyril Auger, Anne-Laure Charles, Samy Talha, Alain Meyer, Francois Piquard, Emmanuel Andres, Valerie Schini-Kerth, Bernard Geny

**Affiliations:** 1Fédération de Médecine Translationnelle de Strasbourg (FMTS), Faculté de Médecine, Université de Strasbourg, Equipe D’accueil 3072, 11 Rue Humann, 67000 Strasbourg, France; Olivier.rouyer@chru-strasbourg.fr (O.R.); anne.laure.charles@unistra.fr (A.-L.C.); samy.talha@chru-strasbourg.fr (S.T.); alain.meyer1@chru-strasbourg.fr (A.M.); francois.piquard@wanadoo.fr (F.P.); 2Service de Physiologie et D’explorations Fonctionnelles, Hôpitaux Universitaires de Strasbourg, Nouvel Hôpital Civil, 1 Place de l’Hôpital, 67091 Strasbourg CEDEX, France; 3CNRS UMR 7213, Laboratoire de Biophotonique et Pharmacologie, Faculté de Pharmacie, Université de Strasbourg, 67401 Illkirch, France; cyril.auger@unistra.fr (C.A.); valerie.schini-kerth@pharma.u-strasbg.fr (V.S.-K.); 4Service de Médecine Interne, Diabète et Maladies Métaboliques, Pôle M.I.R.N.E.D., Hôpitaux Universitaires, CHRU Strasbourg, 67000 CEDEX, France; Emmanuel.andres@chru-strasbourg.fr

**Keywords:** high-fat meal, flow-mediated dilatation, endothelial function, hypertriglyceridemia, inflammation, oxidative stress, glycemia, polyphenols, juice, champagne

## Abstract

Endothelial dysfunction (ED), often linked to hypertriglyceridemia, is an early step of atherosclerosis. We investigated, in a randomized cross-over study, whether high-fat meal (HFM)-induced ED might be reduced by fruit juice or champagne containing polyphenols. Flow-mediated dilatation (FMD) and biological parameters (lipid profile, glycemia, inflammation, and oxidative stress markers) were determined before and two and three hours after the HFM in 17 healthy young subjects (24.6 ± 0.9 years) drinking water, juice, or champagne. Considering the entire group, despite significant hypertriglyceridemia (from 0.77 ± 0.07 to 1.41 ± 0.18 mmol/L, *p* < 0.001) and a decrease in Low Density Lipoprotein (LDL), the FMD was not impaired. However, the FMD decreased in 10 subjects (from 10.73 ± 0.95 to 8.13 ± 0.86 and 8.07 ± 1.16%; *p* < 0.05 and *p* < 0.01; 2 and 3 h, respectively, after the HFM), without concomitant change in concentration reactive protein or reactive oxygen species, but with an increase in glycemia. In the same subjects, the FMD did not decrease when drinking juice or champagne. In conclusion, HFM can impair the endothelial function in healthy young subjects. Fruit juice, rich in anthocyanins and procyanidins, or champagne, rich in simple phenolic acids, might reduce such alterations, but further studies are needed to determine the underlying mechanisms, likely involving polyphenols.

## 1. Introduction

It is now well established in animals and humans that endothelial cells have a decisive role in the control of vascular homeostasis. Their protective effect is explained by endothelial cells’ ability to release powerful vasoactive factors, and to inhibit the proliferation and migration of vascular smooth muscle cells and the expression of many pro-atherothrombotic molecules. Nitric oxide (NO) is one of the most powerful of these vasoactive factors, and NO secretion in response to increased shear-stress is known to mediate flow-mediated dilatation (FMD). Endothelial dysfunction, characterized by a reduction in endothelial-dependent relaxations, is a first step evoking atherosclerosis. Endothelial dysfunction is observed early in cases of high blood pressure, hyperlipemia, and diabetes, cardiovascular risk factors that often precede atherothrombosis and major cardiovascular diseases [1,2,3,4,5]. The endothelial function is now investigated routinely in normal subjects and in patients through the humeral flow-mediated dilatation arising in response to acute ischemia [5,6,7,8,9].

Some clinical studies indicate that impaired endothelial function can be observed in healthy subjects following consumption of a high-fat meal (50–105 g), but not after a low-fat meal (less than 10 g) [10,11]. Impaired endothelial function is often observed 2–4 h after dietary intake and is associated with peak postprandial hyperlipemia [12]. It has been suggested that postprandial hypertriglyceridemia may contribute, at least in part, to such endothelial dysfunction via increased oxidative stress [13]. Very recently, Durrer C. et al. also observed a reduced FMD after HFD in healthy young men [14].

On the other hand, numerous experimental studies suggest that natural products derived from grapes have a beneficial effect on endothelial function in physiological and pathophysiological conditions. Thus, various products derived from grapes, such as skin or seed extract, and certain grape juices or wines are capable of inducing endothelial-dependent relaxation by stimulating the endothelial formation of the powerful vasoprotective factors NO and Endothelium-Derived Hyperpolarizing Factor (EDHF). In addition, regular consumption of a polyphenolic extract of red wine prevented high blood pressure and endothelial dysfunction induced by angiotensin II infusion in rats [15,16]. Accordingly, polyphenols have been shown to counteract the effects of aging on endothelial and mitochondrial function allowing maintained physical performance [17,18].

Clinical studies also indicated that the consumption of products rich in polyphenols, such as grape derivatives and dark chocolate, have a beneficial effect on endothelial function. For example, the consumption of red wine and alcohol-free red wine increased FMD in healthy subjects [19]. FMD was also improved by consuming 4 mL/kg of acute red wine (pinot noir) or white wine (sauvignon blanc), and 7.7 mL/kg of grape juice for 14 days in coronary patients [20,21]. The beneficial effect on endothelial function was greatest about two hours after ingestion of polyphenols [22]. Red wine is an important source of polyphenols (about 1–4 g/L) compared to white wine (about 400 mg/L). In the case of red wine, the beneficial effect has been attributed mainly to flavonoids from the anthocyanins (which give color to the wine), procyanidins (catechin epicatechin complex), polyphenols. In the case of white wine, the beneficial effects have been attributed to its higher content of simple phenolic acids, such as caffeic acid, tyrosol, and shikimic acid, and to a better bioavailability [1,23,24,25].

The aim of this study was, therefore, to determine whether a high-fat meal might impair FMD in young and healthy subjects and whether fruit juice, rich in anthocyanins and procyanidins, or champagne wine, rich in simple phenolic acids, might reduce such alterations in endothelial function, possibly in relation to lipid profile, glycemia, inflammation, and oxidative stress markers changes.

## 2. Population and Methods

### 2.1. Population

Seventeen men, aged between 20 and 35, were enrolled in the study. They were non-smokers, or smoked less than 5 cigarettes per day, and did not take medication. All subjects signed an informed consent form and the study was approved by the ethical committee (Comité de Protection des Personnes EST IV; number 10/51; approved 24/11/2010).

### 2.2. Study Design

This was a randomized, blind, monocentric, cross-over study. In accordance with international recommendations, the patients refrained from strenuous physical activity ≥24 h and fasted for 12 h before the study. Beginning the same hour in the morning to prevent diurnal variation in FMD response, the subjects lay in a quiet room where the temperature was held constant. The high-fat meal consisted of 81 g fat, 101 mg cholesterol, 230 g carbohydrate, and 44 g protein and the subjects’ drank either 300 mL of water, fruit juice, or champagne. Each subject was tested three times, the three sets of the study being separated by a 7-day “wash-out” period.

### 2.3. Parameters Determined

#### 2.3.1. Hemodynamic Parameters

For FMD determination, we measured the diameter of the humeral artery by ultrasonography, according to previous reports [6,7,8,9], using a high-resolution 2-dimensional ultrasound imaging system (ATL HDI 5000; Advanced Technology Laboratories, Bothell, WA, USA) in B-mode. Electrocardiography-triggered ultrasound images were obtained with a high-resolution linear-array transducer (15–7 MHz). Ultrasound parameters were set to optimize longitudinal B-mode images of the lumen/arterial wall interface.

After a resting period >15 min, the probe was fixed and the patients arm remained in the same position throughout the study. Baseline recording of the arterial diameter was performed and ischemia was obtained using an occlusion cuff inflated to 50 mmHg higher than the patient’s baseline systolic blood pressure during 5 min. The same settings were maintained during the study, and FMD was calculated as the largest change in the brachial artery diameter with reperfusion, at the peak of the R wave of the EKG. Diameter was measured at baseline and immediately after cuff deflation, at 20, 40, 60, and 80 s. The FMD, measured before the meal and 2 h (peak bioavailability of polyphenols) and 3 h after eating, was expressed as: (maximal diameter − basal diameter) ÷ (basal diameter)) × 100(1)

Heart rate (HR) and systemic blood pressure were also determined.

#### 2.3.2. Biological Parameters

Blood samples allowed us to determine the kinetic of lipid profile (plasma triglyceride levels, total cholesterol, Low-density lipoprotein (LDL), High-density lipoprotein (HDL), cholesterol), glycemia, and ultrasensitive C reactive protein (CRP) using routine biochemical analysis. Four venous blood samples were drawn before, and 1, 2, and 3 h after the HFM, using a venous line (18 Gauge catheters).

Oxygen reactive species (ROS) production was determined, as previously reported [26]. Briefly, 20 µL of blood was mixed with 20 µL of spin probe CMH (1-hydroxy-3-methoxycarbonyl-2, 2, 5, 5-tetramethylpyrrolidine HCl) and measured at 37 °C by e-scan spectrometer (Bruker, Germany). Detection of ROS production was conducted using BenchTop EPR (electron paramagnetic resonance) spectrometer E-SCAN under the following EPR settings: center field g = 3477.452; field sweep 60 G; microwave power 21.85 mW; modulation amplitude 2.40 G; conversion time 10.24 ms; time constant 40.96 ms; and number of scans: 10. Results were expressed as µmol/min.

### 2.4. Statistical Analysis

All data are expressed as mean ± standard error of the mean (SEM), and were analyzed using Prism software (GraphPad Prism 5, Graph Pad Software, San Diego, CA, USA). We tested all the parameters for normality assumption in all groups, using the Shapiro–Wilk test. When one parameter was not distributed normally, we performed a non-parametric test on repeated values (Friedman test) followed by a post test (Dunn’s multiple test) for the entire data sets (all time points). LDL and glycemia were distributed normally in all groups, and the Bartlett’s test demonstrated they meet the homogeneity assumption. We therefore applied a parametric test on repeated values (ANOVA) followed by a post test (Newman–Keuls test). In all cases, a *p* value <0.05 was considered significant.

## 3. Results

### 3.1. Characteristics of the Subjects

The main clinical and biological characteristics of the 17 subjects are presented in Table 1. Corresponding to the inclusion criteria, they were young and healthy. 

### 3.2. Effects of the High-Fat Meal on the Entire Population Drinking Water

#### 3.2.1. FMD Evolution

At the systemic level, we did not observe any significant variations in heart rate or blood pressure during the protocol.

Before the high-fat meal, the endothelial function of the subjects was in the normal range (9.69% ± 0.86%). The endothelial function after the HFM was not significantly modified (8.69% ± 0.85%, and 8.87% ± 0.88%, 2 and 3 h post-meal, respectively, Figure 1A).

#### 3.2.2. Biological Effects

Concerning the lipid profile, the HFM induced a significant increase in plasma triglyceride levels (from 0.77 ± 0.07 to 1.29 ± 0.15 (*p* < 0.01) and 1.41 ± 0.18 mmol/L (*p* < 0.001), 2 and 3 h post-meal, respectively, Figure 1E). No changes in plasma levels of total and HDL cholesterol were observed (Figure 1F,G), but there was a gradual decrease in plasma LDL cholesterol levels over time (from 2.33 ± 0.15, to 2.17 ± 0.14 (*p* < 0.001) and 2.11 ± 0.15 mmol/L (*p* < 0.001), 2 and 3 h post-meal, respectively) (Figure 1H).

Glycemia did not show significant variations (Figure 1B) and the high-fat meal did not modify the plasma concentration of ultra-sensitive CRP (Figure 1C). Similarly, ROS production showed no difference after the HFM (Figure 1D).

However, since individual responses might differ, we investigated them and, thereby, identified a subgroup of 10 subjects out of the 17, in whom the FMD decreased after the high-fat meal. Their characteristics are presented in Table 2. Particularly, knowing that baseline FMD and glycemia might influence subsequent FMD modulation by meal ingestion, we investigated their possible differences in the three sets of the study in the 10 selected subjects.

Before HFM ingestion, the values of FMD were 10.73 ± 0.95, 8.17 ± 0.92, and 9.45 ± 0.82 in the 10 same subjects while randomly drinking water, juice, and champagne, respectively. Baseline FMD values were significantly lower in the subjects when drinking juice as compared to water (*p* < 0.05). Before HFM ingestion, the values of glycemia were 5.11 ± 0.17, 5.21 ± 0.10, and 5.19 ± 0.08 in the subjects drinking water, juice, and champagne, respectively. No significant difference was observed between any groups.

On the basis of similar published data, demonstrating the interest of a stratification analysis [27], we present the results observed in this selected subgroup (*n* = 10 for all parameters except for EPR, *n* = 5) when drinking water (Figure 2), juice (Figure 3), or champagne (Figure 4). 

### 3.3. Effects of the High-Fat Meal on the Selected Ten Subjects Showing Decreased FMD When Drinking Water

#### 3.3.1. FMD Evolution

In these selected patients, FMD decreased from 10.73 ± 0.95 to 8.13 ± 0.86 (*p* < 0.01) and 8.07 ± 1.16 (*p* < 0.05) 2 and 3 h, respectively, after the HFM (Figure 2A).

#### 3.3.2. Biological Effects

The triglyceridemia significantly increased from 0.81 ± 0.11 to 1.41 ± 0.22 mmol/L and 1.52 ± 0.27 mmol/L (*p* < 0.01) at 2 and 3 h after the HFM, respectively (Figure 2E). Total cholesterol also increased from 4.23 ± 0.3 to 4.36 ± 0.9 (*p* < 0.05) and 4.33 ± 0.31 mmol/L, (*p* < 0.05), respectively, at 2 and 3 h after the HFM (Figure 2F). HDL levels did not vary significantly (Figure 2G), but LDL levels decreased significantly from 2.43 ± 0.25 to 2.25 ± 0.22 mmol/L (*p* < 0.05) and 2.19 ± 0.24 mmol/L (*p* < 0.01) at 2 and 3 h after the HFM, respectively (Figure 2H).

The HFM also increased the glycemia from 5.11 ± 0.16 to 5.62 ± 0.23 mmol/L (*p* < 0.05) and to 5.81 ± 0.36 mmol/L (*p* < 0.01), at 2 and 3 h after the meal, respectively (Figure 2B). usCRP (Figure 2C) and ROS production (Figure 2D) were not modified.

### 3.4. Effects of the High-Fat Meal on the Selected Subjects Drinking the Fruit Juice

#### 3.4.1. FMD Evolution

With juice ingestion, the high-fat meal did not modify the endothelial function and, thus, FMD was not reduced (Figure 3A).

#### 3.4.2. Biological Effects

The HFM did not affect any biological parameters in the subgroup except triglyceridemia, which increased from 0.77 ± 0.11 mmol/L to 1.33 ± 0.19 mmol/L (*p* < 0.05) and to 1.28 ± 0.20 mmol/L (*p* < 0.01), at the second and third hour after the HFM, respectively (Figure 3E). Further, the LDL levels were significantly decreased (2.49 ± 0.21 mmol/L before the HFM, and 2.24 ± 0.22 mmol/L (*p* < 0.001) and 2.29 ± 0.22 mmol/L, (*p* < 0.001) 2 and 3 h after the HFM, respectively (Figure 3H).

### 3.5. Effects of the High-Fat Meal on the Selected Patients Drinking Champagne

#### 3.5.1. FMD Evolution

The high-fat meal did not modify significantly the FMD in the subject group drinking champagne (Figure 4A).

#### 3.5.2. Biological Effects

Again, the HFM did not affect any biological parameters in the selected subgroup of the 10 patients except for triglyceridemia, which increased significantly after the HFM from 0.96 ± 0.19 mmol/L to 1.53 ± 0.23 mmol/L and to 1.85 ± 0.3 mmol/L (*p* < 0.001), at 2 and 3 h after the HFM, respectively. LDL levels were significantly decreased (2.36 ± 0.22 mmol/L before HFM ingestion, and 2.12 ± 0.19 mmol/L (*p* < 0.01) and 2.0 ± 0.2 mmol/L (*p* < 0.001) after 2 and 3 h, respectively (Figure 4H).

## 4. Discussion

The main findings of this study are that the high-fat meal significantly and similarly increased triglyceridemia in the three experimental sets, and that 10 out of the 17 subjects demonstrated a significant FMD decrease when drinking water, but not when drinking either fruit juice or champagne.

### 4.1. Effects of the High-Fat Meal When Drinking Water

Considering the entire group of 17 subjects, the HFM did not result in significant FMD changes despite the increase in triglyceridemia, which is thought to be a major causal factor of endothelial dysfunction [12,13]. A greater FMD decrease might have been observed in a population characterized by cardiovascular risk factors, but we found it to be of interest to assess the potential deleterious effects of this type of meal often eaten by young subjects in whom atherosclerosis and cardiovascular disease might occur later on. Indeed, endothelial dysfunction is interesting to investigate in young subjects since it represent a very early event in the atherosclerosis process.

As shown recently [14], individual variations might occur. We therefore analyzed each subject’s responses to the HFM and, interestingly, the FMD decreased significantly in 10 subjects. We will now specifically discuss the data obtained in these 10 subjects characterized by a decrease in FMD after the HFM. Endothelial dysfunction was observed after short-term HFM in experimental animals [28]. Further, FMD has been shown to be reduced after a single HFM in human [29]. Besides the increase in triglycerides and total cholesterol that likely explain the decreased FMD, other parameters, such as oxidative stress, inflammation, and acute hyperglycemia, might be involved [13,30,31,32,33,34,35]. In our study, the high-fat meal did not induce changes in short-term oxidative stress, as inferred from ROS production determination and the inflammatory status of the subjects—assessed by the plasma ultra-sensitive CRP assay—was not modified. However, acute hyperglycemia is well-known to induce endothelial dysfunction [31,32,33] and although we cannot infer from our results a causal relationship, it is of note that glycemia increased significantly, albeit lightly, in these subjects. Concerning the significant LDL decrease after the HFM, there are few data in the literature. Liu et al. observed similar LDL values after peanut consumption [27]. On the other hand, Dalbani et al. reported that decreased LDL after a fat meal was not associated with a decrease in FMD in male healthy subjects eating tomato paste [36].

### 4.2. Effects of Fruit Juice and Champagne Wine on the Postprandial Endothelial Function after the High-Fat Meal

We choose these beverages in view of their differences in polyphenol content. Fruit juice included red fruits, such as grapes, lingonberry, blueberry, strawberry, and *black aronia*. It contained a high concentration of total polyphenols with 3.2 grams of gallic acid equivalent per liter of juice (g/L GAE (Gallic acid equivalent)), measured using the Folin–Ciocalteu reagent according to the Singleton method [37]. Due to its fruit composition, it also overwhelmingly contained anthocyanidins and procyanidins, although other classes of compounds were present, such as flavan-3-ols or phenolic acids [38].

Champagne is made from a blend of various grape varieties (Chardonnay, Pinot Noir, and Pinot Meunier). The content of total polyphenols in Champagne wines is between 200 and 300 mg gallic acid equivalent per liter, which is in high concentrations for white wines (50 to 300 mg/L GAE), but remains significantly lower than red wines (0.8 to 4 g/L GAE). Champagne wines mainly contain phenolic acids, particularly caftaric acid as well as some flavonoids [39].

Thus, our study design allowed us to compare the potential effect of different polyphenols types on HFM-induced impairment in FMD. Interestingly, contrary to the reduced FMD observed when drinking water, no significant modification in FMD was observed in the same subjects drinking either juice or champagne. Important biological parameters, such as inflammatory and oxidative stress markers, did not vary between groups and might not be involved in these results. On the contrary, glycemia increased significantly only when the subjects drank water and not juice or champagne. This was not expected and, although glycemia values were similar in the three sets of the study, this might potentially participate in the greater decrease in FMD observed in the 10 subjects when drinking water. Further, based on literature data, it is likely that the polyphenols present in both drinks might have participated in the lack of FMD decrease since they are known to protect the vascular function both in experimental animals and in humans. Alternatively, considering the juice, baseline FMD might have played a role. Indeed, meal-induced decrease in FMD can be reduced in subjects with significantly lower baseline FMD [40].

### 4.3. Limitations of the Study

Although the clinical relevance of the HFM-associated decrease in FMD observed in the 10 subjects drinking water might be discussed, it appeared that a similar reduction (about 2%) in post-prandial FMD might correspond to an 18% increase in cardiovascular events [41]. Additionally, as stated before, these results were obtained in a relatively small sample size of healthy young subjects and further studies will be useful in a larger sample to determine whether the FMD decrease would be greater in subjects without and with cardiovascular risk factors and whether such beverages containing polyphenols might present with protective effects.

## 5. Conclusions

This study supported that HFM can impair the endothelial function in healthy young subjects. The decrease in FMD was not observed when the subjects drank fruit juice, rich in anthocyanins and procyanidins, or champagne, rich in simple phenolic acids. Further studies will be useful to confirm these data in a larger population and to determine the mechanisms involved. The responses observed in patients demonstrating cardiovascular risk factors and, therefore, potential greater endothelial dysfunctions after an HFM might also be interesting to investigate.

## Figures and Tables

**Figure 1 jcm-08-00859-f001:**
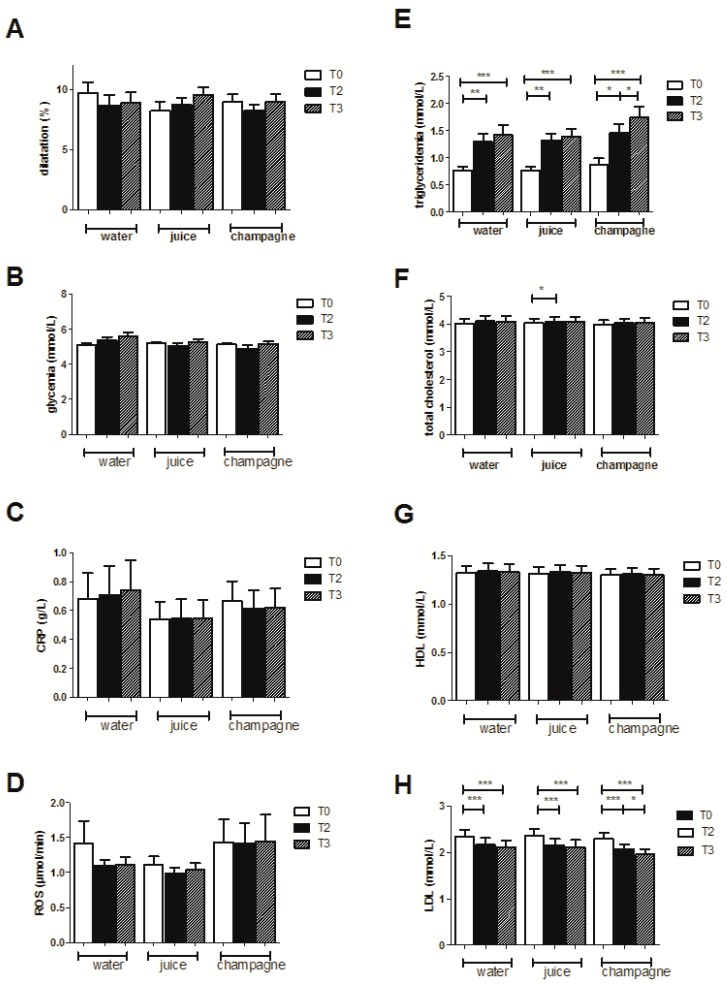
Effects of the high-fat meal and ingestion of water, fruit juice, or champagne in the 17 subjects. (**A**) Endothelial function as quantified by flow-mediated dilatation of the humeral artery (FMD), (**B**) glycemia levels, (**C**) CRP, (**D**) ROS production, (**E**) triglyceridemia, (**F**) total cholesterol, (**G**) LDL, and (**H**) HDL levels. Results are expressed as means ± SEM. * *p* < 0.05, ** *p* < 0.01, *** *p* < 0.001.

**Figure 2 jcm-08-00859-f002:**
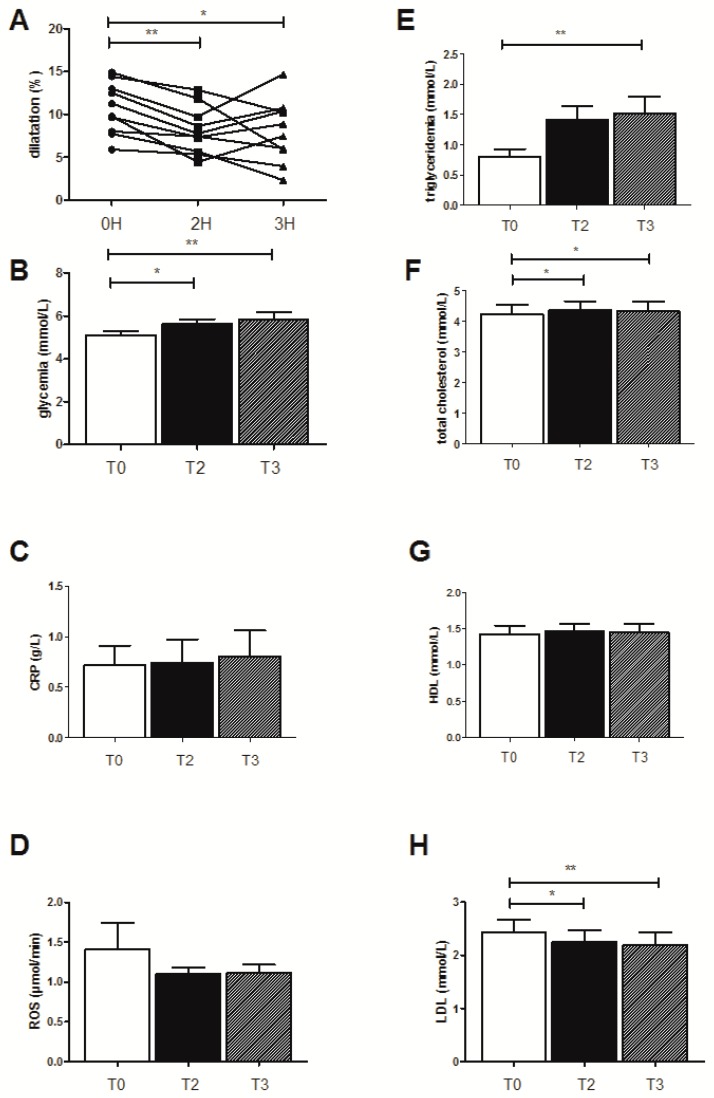
Effects of the high-fat meal on the ten subjects showing decreased FMD when drinking water. (**A**) Endothelial function as quantified by flow-mediated dilatation of the humeral artery (FMD), (**B**) glycemia levels, (**C**) CRP, (**D**) ROS production, (**E**) triglyceridemia, (**F**) total cholesterol, (**G**) LDL, and (**H**) HDL levels. Results are expressed as means ± SEM. * *p* < 0.05, ** *p* < 0.01.

**Figure 3 jcm-08-00859-f003:**
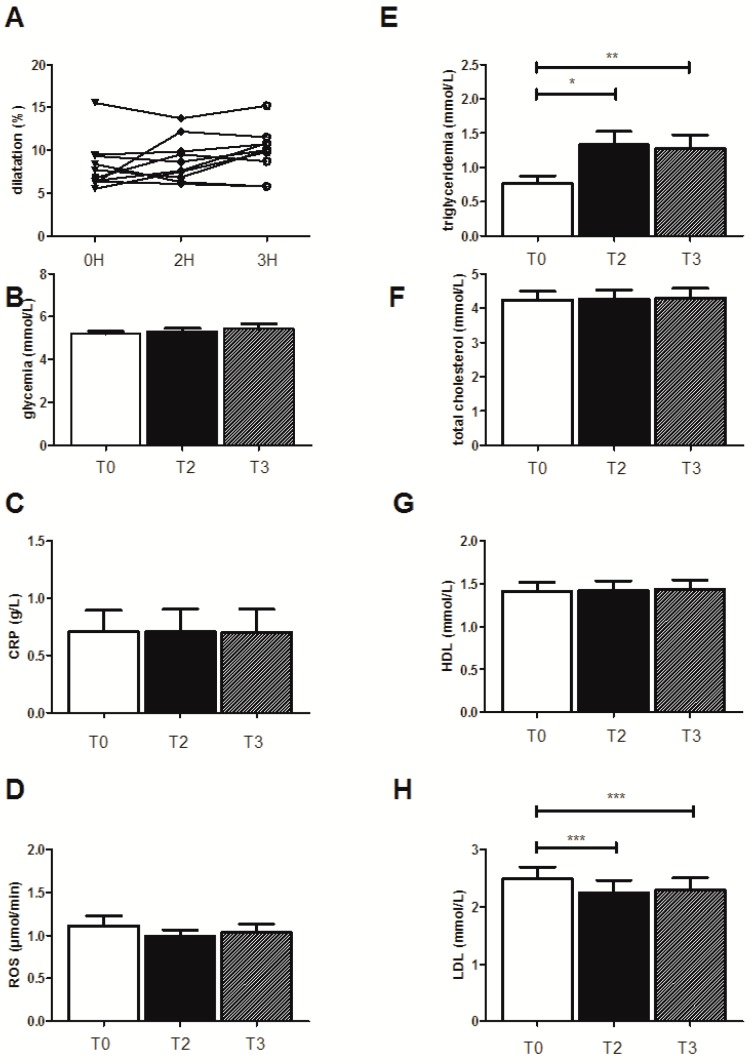
Effects of the high-fat meal on the ten subjects when drinking juice. (**A**) Endothelial function as quantified by flow-mediated dilatation of the humeral artery (FMD), (**B**) glycemia levels, (**C**) CRP, (**D**) ROS production, (**E**) triglyceridemia, (**F**) total cholesterol, (**G**) LDL, and (**H**) HDL levels. Results are expressed as means ± SEM. * *p* < 0.05, ** *p* < 0.01, *** *p* < 0.001.

**Figure 4 jcm-08-00859-f004:**
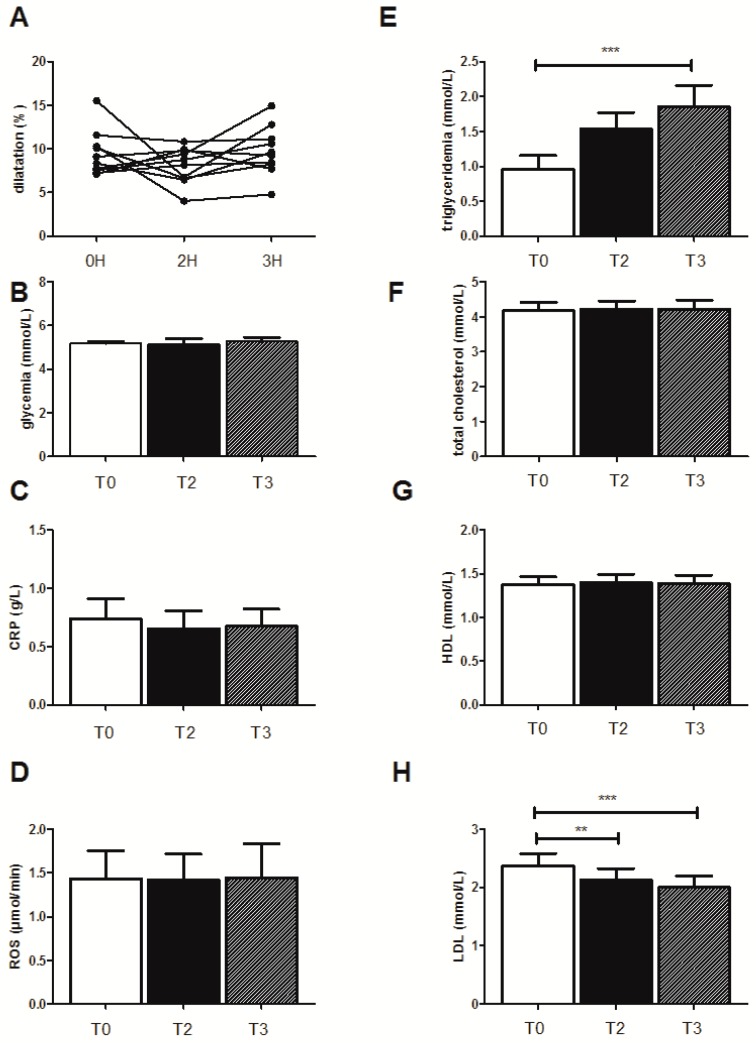
Effect of the high-fat meal on the ten subjects when drinking champagne. (**A**) Endothelial function as quantified by flow-mediated dilatation of the humeral artery (FMD), (**B**) glycemia levels, (**C**) CRP, (**D**) ROS production, (**E**) triglyceridemia, (**F**) total cholesterol, (**G**) LDL, and (**H**) HDL levels. Results are expressed as means ± SEM. ** *p* < 0.01, *** *p* < 0.001.

**Table 1 jcm-08-00859-t001:** Clinical and biological characteristics of the subjects.

**Clinical Characteristics**	
Age (years)	24.6 ± 0.9
Weight (kg)	76.2 ± 2.3
BMI (kg/m^2^)	23.6 ± 0.8
MAP (mmHg)	95.8 ± 1.7
HR (beats/min)	68 ± 2
FMD (%)	9.69 ± 0.86
**Biological characteristics**	
Triglyceridemia (mmol/L)	0.76 ± 0.07
Total cholesterol (mmol/L)	4.01 ± 0.19
HDL (mmol/L)	1.32 ± 0.07
LDL (mmol/L)	2.33 ± 0.15
Glycemia (mmol/L)	5.12 ± 0.11
usCRP (g/L)	0.68 ± 0.18
ROS (µmol/min)	1.42 ± 0.32

BMI: body mass index, MAP: mean arterial systemic blood pressure, HR: heart rate, FMD: flow-mediated dilatation, HDL: High-density lipoprotein, LDL: Low-density lipoprotein, usCRP: ultra-sensitive C reactive protein, and ROS: reactive oxygen species. Mean ± standard error (*n* = 17 for all parameters except for ROS *n* = 11).

**Table 2 jcm-08-00859-t002:** Clinical and biological characteristics of the 10 and 7 selected subjects.

Biological Parameters	10 Patient-Group			7 Patient-Group		
	Water	Juice	Champagne	Water	Juice	Champagne
MAP (mmHg)	95.3 ± 2.3			96.3 ± 2.3		
HR (beats/min)	67.5± 2.4			73.7 ± 2.5		
FMD (%)	10.73 ± 0.95	8.17 ± 0.92 *	9.45 ± 0.82	8.22 ± 1.49	8.31 ± 1.46	8.28 ± 0.98
Triglyceridemia (mmol/L)	0.81 ± 0.11	0.77 ± 0.11	0.96 ± 0.2	0.70 ± 0.06	0.78 ±0.06	0.73 ± 0.09
Total cholesterol (mmol/L)	4.23 ± 0.3	4.24 ± 0.26	4.18 ± 0.23	3.69 ± 0.11	3.74 ± 0.11	3.71 ± 0.08
HDL (mmol/L)	1.43 ± 0.11	1.41 ± 0.1	1.38 ± 0.09	1.18 ±0.05	1.19 ± 0.05	1.19 ± 0.06
LDL (mmol/L)	2.43 ± 0.25	2.49 ± 0.21	2.36 ± 0.22	2.19 ± 0.13	2.20 ± 0.16	2.19 ± 0.09
Glycemia (mmol/L)	5.11 ± 0.17	5.21 ± 0.1	5.19 ± 0.08	5.13 ± 0.12	5.14 ± 0.14	5.09 ± 0.11
usCRP (g/L)	0.72 ± 0.2	0.71 ± 0.18	0.74 ± 0.17	0.63 ± 0.35	0.3 ± 0.08	0.56 ± 0.24
ROS (µmol/min)	1.42 ± 0.32	1.11 ± 0.11	1.43 ± 0.32	0.85 ± 0.13	1 ± 0.07	0.99 ± 0.13

MAP: mean arterial systemic blood pressure, HR: heart rate, FMD: flow-mediated dilatation, HDL: high-density lipoprotein, LDL: low-density lipoprotein, usCRP: ultra-sensitive C reactive protein, and ROS: reactive oxygen species. Results are expressed as means ± SEM. * *p* < 0.05.

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
