# Peer review of "Effects of a High Fat Meal Associated with Water, Juice, or Champagne Consumption on Endothelial Function and Markers of Oxidative Stress and Inflammation in Young, Healthy Subjects"

_jcm, 2019, doi:10.3390/jcm8060859_

Reviewer 1 Report

1. The introduction provides a generalized background of the topic and the authors provides references to other groups who do or have done research in this area. On the other hand the motivations, as well as the objective for this study is clear defined. 

2. The study design is well-defined and in general, the authors provided sufficient information for a capable research to reproduce the experiments described in the manuscript. Nevertheless, for use one way ANOVA, first the authors should be test for normality and homogeneity assumptions. If the data do not meet one of these two assumptions or both, a non-parametric test should be take.

3. According with the results section, the way the author represent the data is clear, however after first peace of data biological effects, they select 10 among 17 original subject, based on that in these subjects the FMD decreased after the high-fat meal. The authors should be provided clinical and biological characteristics for those 10 subjects and compare then with the others 7, especially for basal FMD.

4. Since the number of the sample is small, conclusions should be more moderate

Minor wording changes:

 1.    Page 4, lines 149-154. To be consistent and clear, the panels of figure 1 corresponding to this paragraph could be cited immediately after the respective result, as they do in the following paragraph for glycaemia, CRP and ROS data.

Author Response

Dear Referee 1,

Thank you very much for your nice comments. We revised the manuscript extensively in accord with your suggestions, and answered to your questions. Particularly, we performed first normality and then if needed homogeneity tests. The statistics are not fundamentally changed, some p values were reduced but the results and thus the message are the same. The manuscript is significantly improved with this more adequate analysis and the new table; and we thank you for this improvement.

Please find below the specific changes performed.

With our best regards,

B Geny for the authors

 1.      The introduction provides a generalized background of the topic and the authors provides references to other groups who do or have done research in this area. On the other hand the motivations, as well as the objective for this study is clear defined. 

Thank you very much.

2. The study design is well-defined and in general, the authors provided sufficient information for a capable research to reproduce the experiments described in the manuscript. Nevertheless, for use one way ANOVA, first the authors should be test for normality and homogeneity assumptions. If the data do not meet one of these two assumptions or both, a non-parametric test should be take.

Thank you for this idea. We tested for normality assumption all parameters in all groups. When one parameter time point was not distributed normally, we performed a non-parametric test for the entire data sets (all time points). This was the case for all data except for LDL and glycaemia in all groups that we tested then for homogeneity assumptions. Homogeneity was observed; therefore we kept the parametric test for LDL and glycaemia data, as described in the first draft of the paper.

In the manuscript, both figures are visible. The old one is the first and the second one is the new with the more adequate statistical analysis. Accordingly, the statistical analysis and results paragraph were adapted following this statistical analysis.

As you can see, few change raised, some significance were lower in terms of p value. Thus, the paper message did not change but thank you very much for pushing us to perform a better statistical analysis.

“We tested for normality assumption all parameters in all groups, using the Shapiro-wilk test. When one parameter was not distributed normally, we performed a non-parametric test on repeated values (Friedman test) followed by a post test (Dunn's multiple test) for the entire data sets (all time points). LDL and glycaemia were distributed normally in all groups and the Bartlett’s test demonstrated they meet the homogeneity assumption. We therefore applied a parametric test on repeated values (ANOVA) followed by a post test (Newman Keuls test)”.

3. According with the results section, the way the author represent the data is clear, however after first peace of data biological effects, they select 10 among 17 original subject, based on that in these subjects the FMD decreased after the high-fat meal. The authors should be provided clinical and biological characteristics for those 10 subjects and compare then with the others 7, especially for basal FMD.

Yes, done. Please find below the new table, in the manuscript. There was no significant difference between both groups. Inside the 10 subjects, only baseline FMD in juice was lower than that of water as previously observed.

Table 2. clinical and biological characteristics of the 10 and 7 selected subjects.

10 patient-group

7 patient-group

water

juice

champagne

water

juice

champagne

MAP (mm Hg)

95.3± 2.3

96.3 2.3

HR (beats/min)

67.5± 2.4

73.7 ± 2.5

FMD (%)

10.73±    0.95

8.17 ± 0.92 *

9.45 ±0.82

8.22±    1.49

8.31±    1.46

8.28 ± 0.98

Triglyceridemia   (mmol/L)

0.81 ±0.11

0.77 ±0.11

0.96 ±0.20

0.70 ±0.06

0.78 ±0.06

0.73 ±0.09

Total   cholesterol (mmol/L)

4.23 ± 0.30

4.24 ±0.26

4.18 ±0.23

3.69 ± 0.11

3.74 ± 0.11

3.71 ± 0.08

HDL (mmol/L)

1.43 ± 0.11

1.41 ±0.10

1.38 ±0.09

1.18 ±0.05

1.19 ±0.05

1.19 ±0.06

LDL (mmol/L)

2.43 ±0.25

2.49 ±0.21

2.36 ±0.22

2.19 ±0.13

2.20 ±0.16

2.19 ±0.09

glycemia   (mmol/L)

5.11 ± 0.17

5.21 ±0.10

5.19 ± 0.08

5.13 ±0.12

5.14 ±0.14

5.09 ±0.11

usCRP (g/L)

0.72 ±0.20

0.71 ±0.18

0.74 ±0.17

0.63 ±0.35

0.30 ±0.08

0.56 ±0.24

ROS  (µmol/min)

1.42 ±0.32

1.11 ±0.11

1.43 ±0.32

0.85 ±0.13

1.00 ±0.07

0.99 ±0.13

MAP: mean arterial systemic blood pressure. HR: heart rate. FMD: flow-mediated dilatation. HDL: High-density lipoprotein. LDL: Low-density lipoprotein. usCRP: ultrasensible C reactive protein. ROS: reactive oxygen species. Results are expressed as means ± SEM. * p < 0.05.

4. Since the number of the sample is small, conclusions should be more moderate

Yes, we moderated the comments both in the limitation of the study paragraph and in the conclusions.

Minor wording changes:

 1.    Page 4, lines 149-154. To be consistent and clear, the panels of figure 1 corresponding to this paragraph could be cited immediately after the respective result, as they do in the following paragraph for glycaemia, CRP and ROS data.

Yes, done.

Reviewer 2 Report

The presented study, Geny et al. is focused on determining the effect of high fat meal combined with water, polyphenol-rich juice or champagne on endothelial function as well as selected hemodynamic and biological parameters including oxidative stress and inflammation markers.

Although the study gives a valuable thoughts about the understanding the effects of diet on cardiovascular system functioning, my major concern regarding the article reflect to the population of cases enrolled in the study. Looking at the standard errors of clinical and biological parameters characterizing all the 17 subjects it seems that the group is very homogeneous. However most of the analysis is performed on 10 subject only, which in my opinion is not enough. If the study was based on at least 50 cases, the title could be modified into the direction informing what kind of effect may be induced in cardiovascular system by high fat diet meals, otherwise it is more like a preliminary data.

The introduction is coherent and well brings into the topic of the paper. The methods for validation the hypothesis are properly chosen and well described.

I do not recommend the publication of the paper in the current form.

Author Response

Dear Referee 2,

Thank you very much for your nice comments. Globally you find the paper interesting and of good quality but also, that at least 50 subjects should have been included.

We agree that it might have been better to have a larger population and acknowledge this in both the limitation paragraph and the conclusions of the study.

However, we believe these data are still interesting (particularly using a cross over design) and the number was validated by the methodologist of our University. Accordingly, many published data on this issue were obtained in relatively small population (To confirm our feeling, we checked the number of patients included in 12 published studies we cite and if three of them included more patients then us (n=19, 26 and 58), all the other included less patients than us (n=7,9,10,1011,11,15,15,16). Their results are nevertheless interesting and important to share in order to increase our knowledge on such important issue.

Further, we performed non parametric statistical test where adequate and although as expected some significance were reduced, the overall results and the message did not change.

Thus, acknowledging the limitation you point on, we believe these data deserve to be shared with others, helping thus to develop further research on this important issue.

Hoping you will agree,

With our best regards,

B Geny for the authors

Round  2

Reviewer 1 Report

The points raised have been addressed satisfactorily.

Reviewer 2 Report

The comments were addressed by the authors. I recommend publication of the paper in JCM.